# Untargeted Metabolomics and Physicochemical Analysis Revealed the Quality Formation Mechanism in Fermented Milk Inoculated with *Lactobacillus brevis* and *Kluyveromyces marxianus* Isolated from Traditional Fermented Milk

**DOI:** 10.3390/foods12193704

**Published:** 2023-10-09

**Authors:** Xiao-Lin Ao, Yi-Mo Liao, Hai-Yan Kang, Hong-Li Li, Tian He, Li-Kou Zou, Shu-Liang Liu, Shu-Juan Chen, Yong Yang, Xing-Yan Liu

**Affiliations:** College of Food Science, Sichuan Agricultural University, Ya’an 625014, China; 18981924401@163.com (Y.-M.L.); khy4391@163.com (H.-Y.K.); Lhl07072001@163.com (H.-L.L.); m18328376905@163.com (T.H.); lsliang999@163.com (S.-L.L.); yangyong676@163.com (Y.Y.); lxy05@126.com (X.-Y.L.)

**Keywords:** *Lactobacillus brevis*, *Kluyveromyces marxianus*, fermented milk, untargeted metabolomics, physicochemical analysis, mechanism, quality

## Abstract

Traditional fermented milk from the western Sichuan plateau of China has a unique flavor and rich microbial diversity. This study explored the quality formation mechanism in fermented milk inoculated with *Lactobacillus brevis* NZ4 and *Kluyveromyces marxianus* SY11 (MFM), the dominant microorganisms isolated from traditional dairy products in western nan. The results indicated that MFM displayed better overall quality than the milk fermented with *L. brevis* NZ4 (LFM) and *K. marxianus* SY11 (KFM), respectively. MFM exhibited good sensory quality, more organic acid types, more free amino acids and esters, and moderate acidity and ethanol concentrations. Non-targeted metabolomics showed a total of 885 metabolites annotated in the samples, representing 204 differential metabolites between MFM and LFM and 163 between MFM and KFM. MFM displayed higher levels of N-acetyl-L-glutamic acid, cysteinyl serine, glaucarubin, and other substances. The differential metabolites were mainly enriched in pathways such as glycerophospholipid metabolism, arginine biosynthesis, and beta-alanine metabolism. This study speculated that *L. brevis* affected *K. marxianus* growth via its metabolites, while the mixed fermentation of these strains significantly changed the metabolism pathway of flavor-related substances, especially glycerophospholipid metabolism. Furthermore, mixed fermentation modified the flavor and quality of fermented milk by affecting cell growth and metabolic pathways.

## 1. Introduction

Western Sichuan presents unique microbial resources due to its specific geographical and climatic conditions. Naturally fermented yak yogurt contains abundant lactic acid bacteria (LAB) and yeasts, contributing a unique flavor and texture to fermented dairy products on the western Sichuan plateau of China. Screening and utilizing the dominant strains can help develop fermented dairy products with regional characteristics and provide nutritional enrichment.

Fermented dairy products are popular with consumers due to their rich nutritional value and unique flavor. Commercial fermented milk is mainly produced using LAB as a starter, while traditional products like milk dregs and kefir primarily use yeasts in addition to LAB, such as *Kluyveromyces marxianus* and *Saccharomyces cerevisiae* [1,2,3]. Ni [4] studied the yeast diversity in traditionally fermented products in Xinjiang and Qinghai. The results showed that the yeast species and quantity varied in different regions, with *K. unisporus* (49.43%) and *K. marxianus* (26.44%) dominating in Xinjiang, and *P. fermentans* (41.89%), *S. cerevisiae* (27.02%), and *K. marxianus* (24.32%) being the most abundant in Qinghai. *K. marxianus* is widely present in traditional fermented dairy products. Screening and applying these yeasts as starters or co-starters can help develop new dairy products with local characteristics.

In milk, lactose represents the main carbohydrate. When LAB and yeast are simultaneously inoculated into fermented milk, they can utilize lactose metabolites in complementary ways. Therefore, mixed fermentation is widely used in dairy products [5,6]. The interaction between LAB and yeast during the fermentation process is extremely complex and includes the complementation of their metabolites, influence on cell growth, and impact on the population sensing phenomenon [7,8,9,10]. The complementarity between LAB and yeast in metabolic products is the most common. Arakawa [11] found that LAB uses lactose to produce galactose, which is utilized by yeast to produce pyruvate, while LAB and yeast use pyruvate to form lactic acid.

Sun [12] used metabolomics to compare the metabolite changes during single and mixed fermentation with *L. casei* M2-L2 and *S. cerevisiae* M4-Y2, revealing that mixed fermentation produced more long-chain fatty acids. Higher methionine, phenylalanine, tyrosine, and arginine levels were found in the mixed fermentation system, confirming that amino acids might promote LAB proliferation.

Therefore, this study uses quality analysis and metabolomics technology to examine small- and medium-sized molecular metabolite production during LAB and yeast fermentation while analyzing the metabolic pathways involved in the fermentation process using the bioinformatics database [13] and assessing the mechanism behind flavor substance formation. This may provide a basis for accurately regulating the flavor of LAB- and yeast-fermented milk.

## 2. Materials and Methods

### 2.1. Materials and Equipment

*L. brevis* NZ4 (OQ660312) and *K. marxianus* SY11 (MZ144218) were screened from naturally fermented dairy products in western Sichuan, China. Milk was purchased from local pasture, while the UHPLC-Q Exactive HF-X was obtained from Waltham, MA, USA. The pure methanol, acetonitrile, formic acid, and propanol were supplied by Fisher from Waltham, MA, USA.

### 2.2. Sample Preparation

#### 2.2.1. Preparation of the Strains

The activated *L. brevis* NZ4 and *K. marxianus* SY11 strains were inoculated into the milk and fermented at 36 °C and 28 °C, respectively, until set; after which, they were refrigerated at 4 °C as starters.

#### 2.2.2. Preparation of the Fermented Milk

The sterilized milk was inoculated with the starters at a 5% ratio and fermented at 35 °C until set. The fermented milks inoculated with *L. brevis* NZ4, *K. marxianus* SY11, and *L. brevis* NZ4 + *K. marxianus* SY11 (1:1) were labeled LFM, KFM, and MFM, respectively. The samples were refrigerated at 4 °C for subsequent examination.

### 2.3. Sensory Evaluation, Physical and Chemical Indicators, and Viable Counts

Ten panelists were selected and trained to evaluate the tissue status, odor, and taste of the fermented milk. The acidity and ethanol content were determined according to the GB 5009.239-2016 standard of China and a method described by Wang [14], using phenolphthalein as the indicator, titration with 0.1 N NaOH, and calculation. The organic acids content was obtained via HPLC according to the GB 5009.239-2016 standard of China, while amino acids were determined using an amino acid analyzer, reference GB 5009.14-2016. The live LAB was counted according to the GB 4789.35-2016 standard of China, using the dilution coating plate method, and the same method applied to the counting of yeasts, but referenced the GB 4789.15-2016 standard of China.

### 2.4. UHPLC-Q Exactive HF-X Data Acquisition

#### 2.4.1. Sample Pretreatment and Application

Here, 200 mg of the fermented milk and grinding beads was placed in a centrifuge tube; after which, 400 μL of an extraction solution (methanol: water = 4:1 (*v*:*v*)) containing 0.02 mg/mL of internal standard (L-2-chlorophenylalanin) was added for metabolite extraction. The extract was frozen and ground for 6 min (−10 °C, 50 Hz), followed by low-temperature ultrasound extraction for 30 min (5 °C, 40 kHz). The sample was allowed to stand at −20 °C for 30 min and centrifuged for 15 min (4 °C, 13,000 r/min); after which, the supernatant was collected for analysis. The sample metabolites were mixed at equal volumes to prepare a quality control sample (QC).

#### 2.4.2. Liquid Chromatography Conditions

A UHPLC-Q Exactive HF-X system with an HSS T3 chromatographic column (100 mm × 2.1 mm i.d., 1.8 µm) was used to separate a 2 μL sample, which was subjected to mass spectrometric detection. Mobile phase A consisted of 95% water +5% acetonitrile (containing 0.1% formic acid), while mobile phase B comprised 47.5% acetonitrile +47.5% isopropanol +5% water (containing 0.1% formic acid). These procedures were performed using methods described by Meng [15].

#### 2.4.3. Mass Spectrometric Conditions

The mass spectrometric data were collected using a Thermo UHPLC-Q Exactive HF-X Mass Spectrometer equipped with an electrospray ionization (ESI) source operating in either positive or negative ion mode. The optimal conditions included a heating temperature of 425 °C, a capillary temperature of 325 °C, a sheath gas flow rate of 50 arb, an aux gas flow rate of 13 arb, an ion-spray voltage floating (ISVF) of −3500 V in negative mode and 3500 V in positive mode, and a normalized collision energy of 20–40–60 V rolling for MS/MS. The full MS resolution was 60,000, while that of MS/MS was 7500. The data were acquired in Data-Dependent Acquisition (DDA) mode, while detection occurred in a 70–1050 m/z mass range. These procedures were performed using methods described by Fang [16].

#### 2.4.4. Data Processing and Metabolite Annotation

The raw LC–MS data were imported into Progenesis QI metabolomics processing software (Version 2.0, Waters Corporation, Milford, CT, USA) for baseline filtering, peak recognition, integration, retention time correction, and peak alignment. The mass spectrometric information was matched with the HMDB and Metlin databases to obtain the metabolite data. After searching the database, the data matrix was filled with missing values after removing them using an 80% rule. The response intensity of the sample mass spectrum peak was subjected to sum normalization to obtain the data matrix. The variables with a relative standard deviation (RSD) exceeding 30% in the QC samples were deleted, and the data matrix was obtained via log10 logarithmization for subsequent analysis.

The preprocessed matrix file was subjected to differential assessment using R software package ropls (Version 1.6.2) for principal component analysis (PCA) and orthogonal partial least squares discriminant analysis (OPLS-DA). Differential metabolite selection was based on the variable importance in projection (VIP) value obtained via OPLS-DA and the *p*-value of the Student’s *t*-test. VIP > 1 and *p* < 0.05 values indicated differential metabolites, which were annotated using the KEGG database to determine the related pathways. Python (Version 3.10) software package scipy.stats was used for pathway enrichment analysis, and Fisher’s exact test was employed to determine the most relevant biological pathways for experimental processing.

## 3. Results

### 3.1. The Sensory Description, Physicochemical Indicators, and Viable Counts of the Fermented Milk

The sensory description, acidity, ethanol content, and viable LAB and yeast count of the fermented milk are shown in Table 1. The acidity values of all the fermented milk samples were between 70 and 80 °T, which was within the acceptable range for consumers [17]. The acidity of KFM reached 72.34 °T without the participation of LAB, indicating that *K. marxianus* SY11 displayed a strong acid-producing ability. The low alcohol level in the fermented milk, ranging between 1 and 4 g/L, indicated that the yeast provided a mellow flavor to the fermentation system without causing health or other issues due to high alcohol content. The MFM exhibited a LAB level of 10.35 lg CFU/mL and a yeast content of 9.86 lg CFU/mL, both at a higher order of magnitude, showing that the two microorganisms exhibited excellent proliferation in the fermentation system. The sensory description results indicated that KFM exhibited an uneven, thin texture with bubbles, whey precipitation, an obvious alcoholic flavor, and a bland, sour taste. Therefore, *K. marxianus* SY11 alone was unsuitable for milk fermentation. LFM displayed a smooth, thick texture with a distinct sour aroma, showing that *L. brevis* NZ4 was appropriate as a common yogurt starter. MFM showed a smooth, thick texture with micro-bubbles and the moderate alcohol, ester, and sour aroma characteristic of fermented milk in western Sichuan, China. Therefore, combining *K. marxianus* SY11 and *L. brevis* NZ4 as a starter culture is suitable for producing local, uniquely flavored dairy products.

The volatile flavor compounds in the three kinds of fermented milk samples detected via GC–MS are shown in Table 2. A total of thirty-one volatile flavor compounds were identified, including eight acids, five alcohols, nine esters, three alkanes, and six other volatile components. The order of the volatile substance types and quantities in the fermented milk was MFM (16 types) > LMF (14 types) = KFM (14 types). Acids represented the main volatile components in LMF, while KFM contained mostly acids and alcohols, and MFM consisted primarily of acids and esters. These results indicated a higher aroma compound abundance when combining *L. brevis* and *K. marxianus* for milk fermentation, while the acids reacted with the alcohol to increase the ester content.

As shown in Table 3, seven common organic acids were detected in the three kinds of fermented milk. The lactic acid content was highest in MFM, followed by acetic acid, citric acid, succinic acid, oxalic acid, malic acid, and tartaric acid. MFM also displayed the highest organic acid levels. The malic acid content in KFM was 0.28 mg/100 g, significantly lower than the other two fermented milk samples. Some studies have shown that yeast degrades malic acid [18]. In addition, malic acid is an intermediate product of the TCA cycle. Other research has revealed that malic acid is completely metabolized during mixed *L. brevis* and *K. marxianus* fermentation, while acetic acid is produced in large quantities [19], which is consistent with the malic and acetic acids change trend in this study.

The amino acid determination results are shown in Table 4. A total of 17 free amino acids were detected in the three types of fermented milk. The total free amino acid content was significantly higher in MFM and KFM than in LFM and included seven essential amino acids: threonine, valine, methionine, isoleucine, leucine, phenylalanine, and lysine. These amino acids must be obtained from food due to limited synthesis in the human body. Of these, the lysine levels were the highest, reaching 30.15 mg/100 mL in MFM. Lysine is generally higher in animals than cereal proteins and is vital for promoting human growth and development and regulating metabolic balance [20]. Although arginine can be synthesized in the human body, the synthesis rate is slow. Therefore, it is also known as a semi-essential amino acid [21]. The arginine content in the three fermented milk samples ranged from 0.38 mg/100 mL to 0.51 mg/100 mL.

### 3.2. Metabolite Profiles and Data Analysis

#### 3.2.1. Data Quality Control Analysis

The PCA is shown in Figure 1. The tight QC sample clustering on the PCA map indicated the stability and high repeatability of the program. The R^2^X values of the PCA model in positive and negative ion modes were 0.537 and 0.555, respectively, indicating high fitting accuracy. The three experimental samples showed significant clustering in the PCA plots in both positive and negative ion modes, indicating excellent repeatability and minimal differences in the sample group.

#### 3.2.2. Overview of the Metabolites

Metabolomics detected a total of 9384 peaks in both positive and negative ion modes. Of these, 901 metabolites were annotated via the KEGG and HMDB databases. After removing data with missing values > 20% and QC validation RSD values < 30%, 885 metabolites were ultimately annotated, including 627 for positive ions and 258 for negative ions. KEGG annotated a total of 295 metabolic products, mainly including phospholipids (20 types), fatty acids (14 types), carboxylic acids (12 types), and glycerol phosphate choline (12 types). HMDB annotated a total of 667 metabolites, primarily including lipids (282 species), organic acids (136 species), oxygen-containing organic compounds, such as monosaccharides (70 species), and organic heterocyclic compounds, such as nucleic acids, vitamins, antibiotics (69 species), and benzoids (36 species).

### 3.3. Differential Metabolite Analysis

#### 3.3.1. Overview of the Differences between the Samples

PLS-DA was used to analyze the differences between the samples, as shown in Figure 2A. Positive ion mode revealed significant differences between MFM, LFM, and KFM. However, the distance between LFM and KFM was relatively small, possibly due to the substantial differences between MFM and the other two types. As shown in Figure 2C, considerable differences were also evident between MFM and LFM or KFM in negative ion mode, while an intersection was apparent between LFM and KFM. This may be because the differential metabolites produced by KFM and LFM are mostly volatile substances and cannot be fully reflected by HPLC–MS. A permutation test confirmed the PLS-DA model stability (Figure 3B,D). In positive ion mode, PLS-DA produced a cumulative R^2^X value of 0.7177, a cumulative R^2^Y value of 0.991, and a cumulative Q^2^ value of 0.4027. Negative ion mode produced a cumulative R^2^X value of 0.754, a cumulative R^2^Y value of 0.988, and a cumulative Q^2^ value of 0.3492, indicating that the PLS-DA models displayed excellent fitting and high predictive ability.

#### 3.3.2. Differential Metabolite and Pathway Screening

OPLS-DA was used to obtain the VIP and Student’s test *p*-value. VIP values > 1 and *p* < 0.05 indicated differential metabolites, of which 204 were evident between MFM and LFM, 163 between MFM and KFM, and 151 between LFM and KFM. The metabolites related to the fermented milk flavor (including organic acids, amino acids, sugars, and fatty acids) were screened. The 50 most abundant metabolites were selected to prepare the cluster diagram, as shown in Figure 3. The results showed that the flavor-related metabolites of MFM and KFM were similar, while significant differences were evident between those of LFM and KFM.

The relevant metabolites were searched in the KEGG database and enriched in the KEGG pathway. As shown in Figure 4, five metabolic pathways were highly enriched in the test group, including glycerophospholipid metabolism, arginine biosynthesis, beta-alanine metabolism, arginine and proline metabolism, and benzoxazinoid biosynthesis. A glycerophospholipid metabolic pathway map was drawn using Illustrator software (Version CS5) to further analyze the related upstream and downstream metabolites.

As shown in Figure 5, the annotated metabolites annotated in this pathway included 1-Acyl-sn-glyceeo-3-phosphocholine, CDP-diacylglycerol, and phosphatidyl-L-serine. Phosphatidyl-L-serine was significantly upregulated in MFM compared with LFM and KFM, while CDP-diacylglycerol and phosphatidyl-glycerophosphate were substantially higher than in KFM. The 1-Acyl-sn-glyceeo-3-phospholine level in MFM was significantly lower than in LFM and KFM, suggesting it represented a precursor of this metabolic pathway. The phosphate tidylcholine and 1,2-diacyl-sn-glycerol-3P levels were considerably lower in MFM than in KFM. These two substances may play an intermediary role in the metabolic pathway.

## 4. Discussion

Fermented dairy products are highly popular with consumers worldwide. Yeast can change the quality during the fermentation and ripening of these products. Its high protein decomposition ability can produce flavor-affecting precursors as well as alcohol and carbon dioxide that reduce the pH and inhibit harmful bacterial growth [22,23]. Therefore, this study used mixed *L. brevis* and *K. marxianus* fermentation to prepare alcohol-flavored fermented milk.

MFM and LFM displayed high LAB levels at about 10^10^ CFU/mL. The yeast content was significantly higher in MFM than in KFM, while the ethanol level was lower. This may be because the ethanol in MFM is utilized by LAB or esterified with other substances. Some studies have shown that ethanol and lactic acid esterification can reduce the damage to their own cells to a certain extent [24]. The types and relative content of the esters in the volatile MFM flavor compounds were higher than in KFM and LFM, further indicating a higher number of esterification reactions between the acids and alcohols in MFM. In addition, lactic acid has a chelating effect on cations and can reduce their impact on yeast cells [25,26]. LAB and yeast interact directly or indirectly. Wang [27] revealed that *K. marxianus* significantly promoted *L. casei* proliferation, while *L. lactis* substantially encouraged *K. marxianus* growth.

The free amino acid content was significantly higher in MFM and KFM than in LFM. The nutritional factors required for LAB growth were relatively abundant, with many amino acids consumed during fermentation. Moreover, the free amino acid content was low in LFM since its protein hydrolase enzyme system was not as rich as MFM, which exhibited a higher amino acid level due to protein hydrolyzation by *K. marxianus*. Since the amino acids produced by *K. marxianus* were used by *L. brevis* to maintain growth and metabolism, the free amino acid content was slightly lower in MFM than in KFM. The analysis of the basic physical and chemical indexes showed that *L. brevis* and *K. marxianus* eventually improved the quality of fermented milk by promoting cell growth and causing changes in other indexes.

The flavor substance sources in fermented milk are mostly represented by fat, lactose, and protein as precursors, producing amino acids, sulfur-containing amino acids, peptides, lipids, sugars, and other substances under the action of microorganisms. Since these substances further constitute the flavor components in fermented milk [28], amino acids and their derivatives, organic acids, sugars, and fatty acids are important analytical targets. In this study, the overall abundance of the flavor-related metabolites was higher in KFM than in LFM and included N-acetyltryptophan, methionyl-threonine, and cysteinyl-proline, most of which represented amino acid derivatives. Studies have shown that yeast increases the amino acids in the mixed fermentation system to maintain LAB growth and metabolism [29]. Using transcriptome technology, some studies found significant changes in the biosynthesis or degradation and the transport and metabolism of amino acids and other gene pathways in milk fermented with LAB and yeast [30]. From genotype to phenotype, LAB and yeast may interact using amino acids as a medium during the co-fermentation process. In addition, significant changes in the amino acid metabolism pathway were evident in fermented sourdough, indicating that this phenomenon is widespread and not limited to fermented dairy products [31]. Nopaline acid, neryl rhamnosyl-glucoside, oleyl alcohol, sphingosine, 19-hydroxycinnamyl 19-glucoside blumenol CO-[rhamnosyl-(1->6)-glucoside], and other substance levels, which mostly represented glycosides, were higher in LFM than in MFM. The glycosides produced via *L. brevis* metabolism provided the carbon source for *K. marxianus* growth. Kadyan [32] used non-targeted metabolomics to determine the key *L. plantarum* and *K. lactis* metabolites during single and mixed fermentation. The results indicated a lower sugar content in the mixed fermentation system, possibly improving the sugar utilization rate. Liu [33] revealed that mixed *L. paracasei* and *K. marxianus* fermentation significantly changed sucrose metabolism, glycolysis, and other pathways by stimulating sugar utilization by microorganisms. The flavor metabolites in MFM included N-acetyl-L-glutamic acid, cysteine-serine, glaucarubin, maleic acid, and serotonin, with fatty acid metabolism-related substances being the most abundant at a total of 97 types. Tian [34] tested the lipase activity of five yeast products, revealing that *K. maximus* lipase activity was the highest, and speculated that yeast participated in protein and fat metabolism, accelerating decomposition and producing various fatty and amino acids. This represents the main source of many flavor substances in fermented foods [35]. Other studies have shown that mixed milk fermentation using LAB and yeast increased the total fatty acid and free fatty acid content and the degree of protein hydrolysis, which is consistent with the results of this study [36].

The selected flavor-related differential metabolites in the KEGG metabolic pathway showed that glycerophospholipid metabolism displayed the highest enrichment. Eight test group substances were enriched into the pathway. Glycerophospholipid metabolism included two sub-pathways, namely phosphatidylcholine and phosphatidylethanolamine biosynthesis, while the substances in the test group were mainly enriched in phosphatidylethanolamine biosynthesis, including four compounds, as shown in Figure 5, of which phosphatidylserine played the most significant role. Phosphatidylserine is formed by phosphatidylserine synthase with L-serine and CDP-diacylglycerol as precursors and is widely found in bacterial, yeast, and mammalian cell membranes. It represents the primary acidic phospholipid in human cell membranes and usually forms part of the cell membrane lobule. It accounts for about 2–20% of the total phospholipid mass in human plasma and cell membranes [37,38]. Some studies have shown that phosphatidylserine displays excellent anti-inflammatory and anti-depression properties and improves cognitive brain function [39]. Zhang [40] screened 94 lipid-related differential metabolites in sheep’s milk using lipomics. In addition, glycerophospholipid metabolism represents a vital fatty acid metabolism pathway and plays a crucial role in product flavor formation. Zhang [41] measured the flavor changes during cheese fermentation in Inner Mongolia, screening 37 different metabolites and six key metabolic pathways. The metabolic pathway was closely related to lipid catabolism, indicating that the catabolic lipid activity during the cheese production process affected its flavor formation. Chen [42] analyzed the flavor precursor changes in purple-leaf tea during different production processes using a lipomics system. The results revealed alpha-linolenic acid metabolism and glycerolipid metabolism as critical lipid metabolism pathways in the raw tea and subsequent products, respectively.

Of the other metabolic pathways noted in this experiment, arginine biosynthesis, beta-alanine metabolism, arginine and proline metabolism, and benzoxazinoid biosynthesis exhibited a higher abundance, most of which were related to amino acid synthesis and metabolism. Yang [43] revealed that the total amino acid content initially increased when mixing *Lactobacillus sanfranciscensis *(*L. sanfranciscensis*)** and *Saccharomyces cerevisiae *(*S. cerevisiae*)** to ferment sourdough, followed by a decline. It is speculated that the two microorganisms consume a large number of amino acids during the initial fermentation stage, while the lower pH in the dough during the later fermentation stage increases the protease activity in the flour, promoting amino acid degradation. Furthermore, *S. cerevisiae* and *L. sanfranciscensis* were exposed to metabolic activity, such as protein hydrolysis, where the protein was decomposed into amino acids, increasing the amino acid content in the dough [44,45].

In conclusion, mixed *L. brevis* and *K. marxianus* fermentation promotes the metabolism of milk proteins, carbohydrates, and fats. The significant enrichment of differential metabolites in the glycerophospholipid metabolism pathway can explain the mechanism behind the changes in fermented milk quality caused by mixed fermentation.

## 5. Conclusions

This study reveals the quality formation in fermented milk inoculated with *Lactobacillus brevis* NZ4 and *Kluyveromyces marxianus* SY11, the dominant microorganisms isolated from traditional dairy products. MFM displays a better overall quality with good sensory characteristics, more organic acid types, higher levels of free amino acids and volatile substance esters, and lower acidity and ethanol concentrations than LFM and KFM. The differential metabolites in MFM are mainly enriched in pathways such as glycerophospholipid metabolism, arginine biosynthesis, beta-alanine metabolism, arginine, and proline metabolism. The mixed fermentation of these two strains significantly changes the flavor-related substance metabolism pathway. The results will pave the way for industrial production and flavor regulation of local special fermented milk.

## Figures and Tables

**Figure 1 foods-12-03704-f001:**
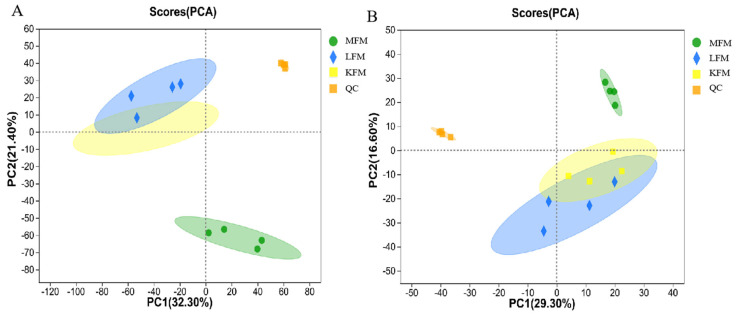
Principal component analysis (PCA) score plots of 3 groups of samples in (**A**) positive [electrospray ionization (ESI+)] (R^2^X = 0.537) and (**B**) negative (ESI−) (R^2^X = 0.555) ion modes. MFM: milk fermented with *L. brevis* NZ4 + *K. marxianus* SY11 (1:1); LFM: milk fermented with *L. brevis* NZ4; KFM: milk fermented with *K. marxianus* SY11; and QC: quality control; the shadow represents the confidence interval.

**Figure 2 foods-12-03704-f002:**
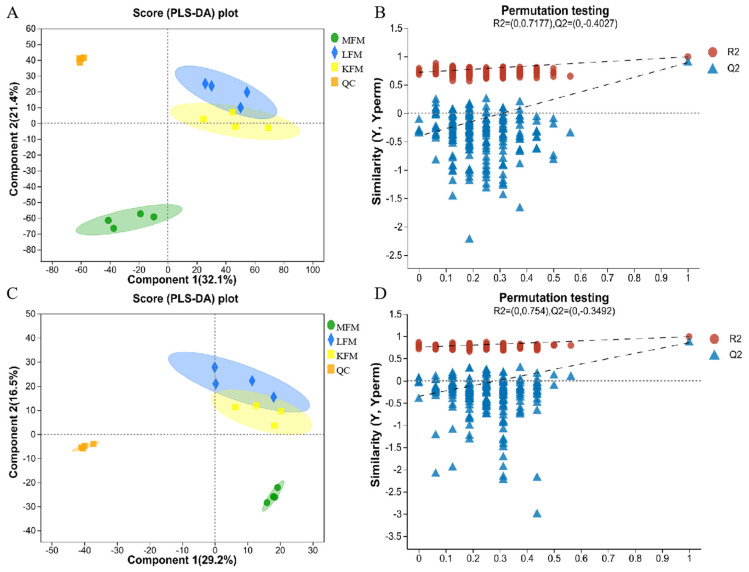
(**A**,**B**) Partial least squares discriminant analysis (PLS-DA) of scores and permutation test plots for the fermented milk analyzed in the positive ion mode. (**C**,**D**) Partial least squares discriminant analysis (PLS-DA) of scores and permutation test plots for the fermented milk analyzed in the negative ion mode; the shadow represents the confidence interval.

**Figure 3 foods-12-03704-f003:**
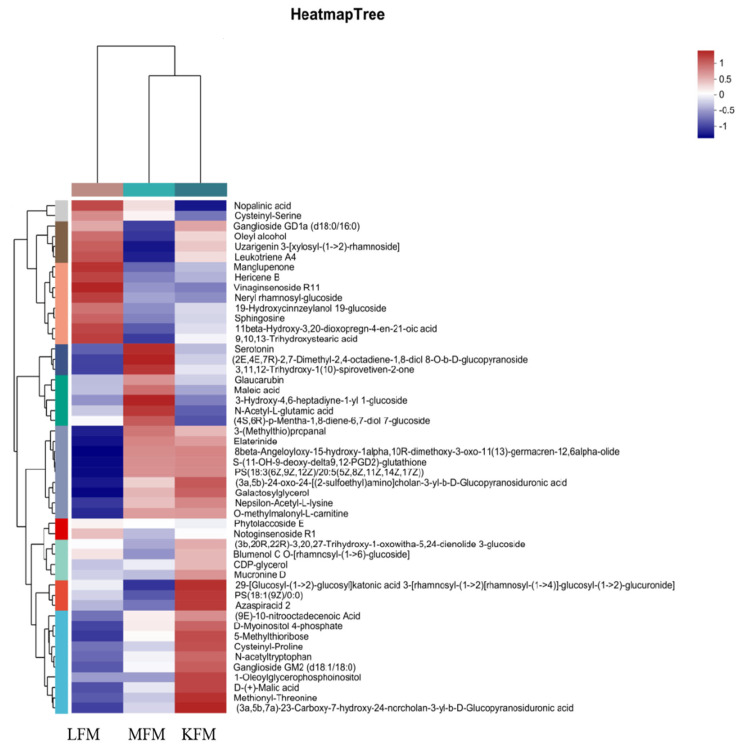
Heatmap of the 50 flavor-related differential metabolites identified in fermented milk samples.

**Figure 4 foods-12-03704-f004:**
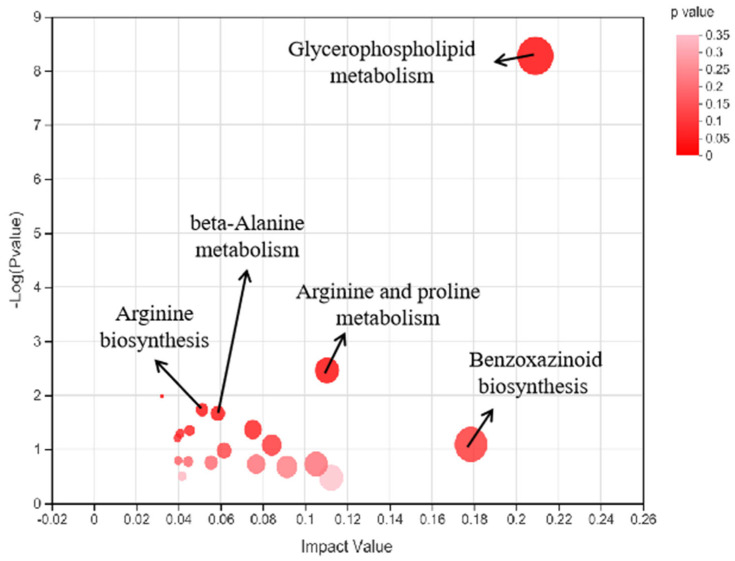
The metabolome view map of the significant metabolic pathways characterized in the fermented milk samples. Circles represent metabolic pathways. The *X*-axis represents the pathway impact, and the *Y*-axis represents pathway enrichment. Larger sizes and darker colors indicate greater pathway enrichment and higher pathway impact values, respectively.

**Figure 5 foods-12-03704-f005:**
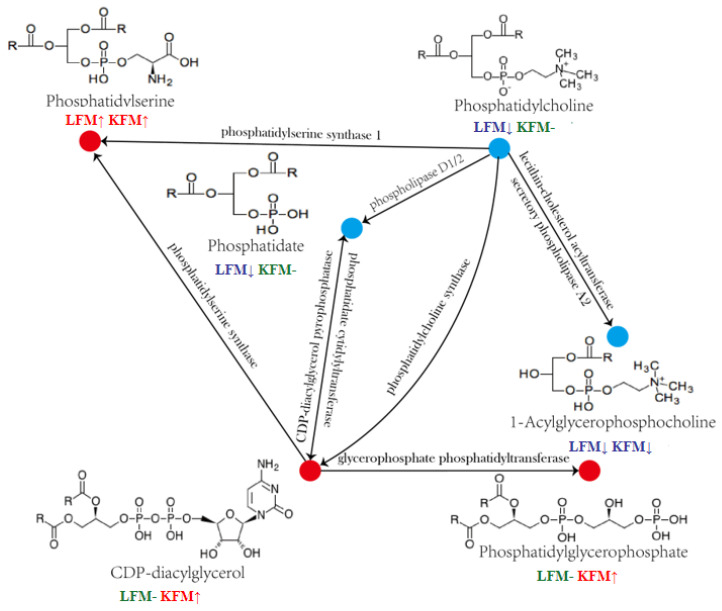
Partial glycerophospholipid metabolic pathway map; red dots represent a relative increase in the content of metabolites, while blue dots represent a relative decrease in the content of metabolites; the upper arrow represents the increase of the metabolite, the lower arrow represents the decrease of the metabolite, and the flat arrow represents no significant change.

**Table 1 foods-12-03704-t001:** Sensory description, physicochemical indicators, and microbial count of the fermented milk.

Item		MFM	LFM	KFM
Acidity (°T)		76.93 ± 0.48 ^a^	78.25 ± 0.59 ^a^	72.34 ± 0.83 ^b^
Ethanol content (g/L)		3.22 ± 0.03 ^b^	1.56 ± 0.12 ^c^	3.99 ± 0.09 ^a^
LAB (lg CFU/mL)		10.35 ± 0.27 ^a^	10.6 ± 0.13 ^a^	—
Yeast (lg CFU/mL)		9.86 ± 0.33 ^a^	—	9.09 ± 0.22 ^b^
Sensory description	Appearance	Thick, no whey	Thick, a small amount of whey	Thin, a small amount of whey
	Texture	Micro-bubble, smooth	No bubble, smooth	Bubble, unsmooth
	Flavor	Moderate alcohol, ester, and sour aroma	Obvious sour aroma	Obvious alcohol flavor and mild acid aroma

Note: MFM: milk fermented with *L. brevis* NZ4 + *K. marxianus* SY11 (1:1); LFM: milk fermented with *L. brevis* NZ4; KFM: milk fermented with *K. marxianus* SY11; and a, b, and c represent the significant differences in the same column, *p* < 0.05.

**Table 2 foods-12-03704-t002:** Relative content of volatile flavor compounds in the fermented milk.

Volatile Substances		Relative Content (%)		
		MFM	LFM	KFM
Acids	Acetic acid	25.16 ± 2.65	17.72 ± 3.49	28.28 ± 1.02
	Butanoic acid	—	6.51 ± 0.72	—
	Pentanoic acid	6.16 ± 0.90	9.10 ± 0.74	12.27 ± 0.68
	n-Decanoic acid	13.72 ± 0.51	11.76 ± 1.49	15.19 ± 2.48
	Octanoic acid	12.21 ± 1.24	11.60 ± 1.40	—
	Propanedioic acid	—	—	6.68 ± 0.52
	Hexanoic acid, 4-methyl-	—	5.29 ± 0.28	—
	6-Hydroxy-2-naphthoic acid	2.57 ± 0.23	—	—
Alcohols	Ethanol	10.33 ± 2.81	—	—
	3-Nonanol, 3-methyl-	7.27 ± 0.76	—	—
	3-Pentanol, 3-methyl-	—	4.02 ± 0.20	—
	1-Propanol, 3-chloro-	—	—	0.28 ± 0.04
	4-Methoxy-4-methyl-2-pentanol	—	—	5.46 ± 0.27
Esters	2-Chloroethyl benzoate	13.18 ± 2.43	10.41 ± 1.13	16.30 ± 2.49
	Butanoic acid, 3-methyl-	—	2.54 ± 1.76	—
	Oxalic acid, cyclobutyl hexyl ester	—	2.20 ± 0.46	—
	1-Propen-2-ol, formate	1.61 ± 0.04	—	—
	Benzeneacetic acid, α-oxo-, methyl ester	5.67 ± 0.57	—	—
	Ethyl hydrogen oxalate	8.40 ± 0.48	—	—
	Ethyl oxamate	10.27 ± 0.57	—	—
	Ethanol, 2-methoxy-, acetate	—	—	3.64 ± 0.57
	2-Propenoic acid, butyl ester	—	—	4.07 ± 0.45
Taxanes	Pentane, 2,3,3,4-tetramethyl-	—	4.59 ± 0.57	—
	Cyclobutane, methoxy-	0.70 ± 0.27	—	—
	3-Pentanamine	—	—	0.79 ± 0.09
Qthers	Acetoin	6.98 ± 2.53	22.77 ± 2.41	13.88 ± 2.42
	Acetone	2.32 ± 0.89	2.14 ± 0.88	3.74 ± 0.24
	2-Propanamine, N-methyl-	—	0.25 ± 0.18	—
	Toluene	0.77 ± 0.17	—	—
	Formic acid hydrazide	—	—	2.50 ± 0.07
	Acetamide, N-methyl-	—	—	1.72 ± 0.13

**Table 3 foods-12-03704-t003:** Organic acid content of the fermented milk.

Type (mg/100 g)	MFM	LFM	KFM
Oxalic acid	17.58 ± 0.18 ^c^	21.44 ± 0.14 ^b^	23.04 ± 0.14 ^a^
Tartaric acid	1.89 ± 0.01 ^c^	2.02 ± 0.004 ^b^	2.52 ± 0.01 ^a^
Malic acid	6.57 ± 0.03 ^b^	8.51 ± 0.18 ^a^	0.28 ± 0.02 ^c^
Lactic acid	224.29 ± 1.67 ^a^	200.05 ± 1.50 ^b^	190.83 ± 0.57 ^c^
Acetic acid	88.50 ± 0.73 ^b^	91.74 ± 0.52 ^a^	92.93 ± 1.53 ^a^
Citric acid	55.88 ± 0.37 ^b^	60.92 ± 0.97 ^a^	59.71 ± 0.41 ^a^
Succinic acid	27.76 ± 1.21 ^a^	25.26 ± 1.10 ^a^	26.82 ± 2.20 ^a^
Total	422.46 ± 3.33 ^a^	409.94 ± 2.08 ^b^	396.13 ± 1.34 ^c^

Note: a, b, and c represent the significant differences in the same column, *p* < 0.05.

**Table 4 foods-12-03704-t004:** Free amino acid content of the fermented milk.

Type	Free Amino Acid Content (mg/100 mL)		
	MFM	LFM	KFM
Aspartic acid	10.81 ± 0.16 ^a^	7.13 ± 0.10 ^c^	9.89 ± 0.29 ^b^
Threonine	2.49 ± 0.12 ^c^	4.10 ± 0.15 ^a^	3.14 ± 0.23 ^b^
Serine	1.37 ± 0.10 ^a^	0.70 ± 0.23 ^b^	1.08 ± 0.09 ^ab^
Glutamic acid	40.22 ± 0.28 ^a^	35.67 ± 0.50 ^b^	38.95 ± 0.87 ^a^
Glycine	6.63 ± 0.04 ^b^	7.03 ± 0.05 ^a^	6.79 ± 0.17 ^ab^
Cysteine	0.61 ± 0.26 ^a^	0.28 ± 0.01 ^a^	0.91 ± 0.37 ^a^
Alanine	0.30 ± 0.03 ^b^	0.64 ± 0.18 ^ab^	0.84 ± 0.15 ^a^
Valine	0.55 ± 0.21 ^b^	0.45 ± 0.01 ^b^	0.90 ± 0.05 ^a^
Methionine	2.45 ± 0.71 ^a^	2.15 ± 1.04 ^a^	3.90 ± 1.10 ^a^
Isoleucine	4.03 ± 0.15 ^b^	4.08 ± 0.20 ^b^	5.09 ± 0.03 ^a^
Leucine	6.11 ± 0.07 ^b^	4.36 ± 0.06 ^c^	6.45 ± 0.08 ^a^
Tyrosine	3.10 ± 1.18 ^a^	3.58 ± 0.21 ^a^	3.90 ± 0.82 ^a^
Phenylalanine	2.69 ± 0.05 ^ab^	2.27 ± 0.05 ^b^	2.98 ± 0.28 ^a^
Lysine	30.15 ± 1.22 ^a^	24.08 ± 0.30 ^b^	28.72 ± 0.61 ^a^
Histidine	2.21 ± 0.15 ^a^	2.15 ± 0.49 ^a^	2.41 ± 0.01 ^a^
Arginine	0.38 ± 0.04 ^a^	0.45 ± 0.17 ^a^	0.51 ± 0.13 ^a^
Proline	9.10 ± 0.23 ^a^	8.89 ± 0.12 ^a^	9.65 ± 0.47 ^a^
Total	123.19 ± 0.62 ^a^	108.00 ± 1.82 ^b^	126.12 ± 3.82 ^a^

Note: a, b, and c represent the significant differences in the same column, *p* < 0.05.

## Data Availability

The data presented in this study are available in the article.

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
