# Peer review of "Untargeted Metabolomics and Physicochemical Analysis Revealed the Quality Formation Mechanism in Fermented Milk Inoculated with Lactobacillus brevis and Kluyveromyces marxianus Isolated from Traditional Fermented Milk"

_foods, 2023, doi:10.3390/foods12193704_

Round 1

Reviewer 1 Report

Study results show interesting metabolite profiles, with a wide collection of data including the influence on growth by different metabolites. Adding some more finale comments in Discussion Chapter could increase the effectiveness of the whole study

The work is written in good quality language, an overall further control (also for syntaxi in a very few cases) could help to obtain further better results

Author Response

Please refer to Appendix.  

Reviewer 2 Report

The authors should conclude and provide proper recommendation for the research work done. who are the final beneficiaries of this research?

Author Response

Please refer to Appendix.

Reviewer 3 Report

The manuscript entitled “Untargeted metabolomics and physicochemical analysis re- vealed the quality formation mechanism in fermented milk in oculated with Lactobacillus brevis and Kluyveromyces marxianus isolated from traditional fermented milk” summarized the molecular mechanism and quality formation mechanism in fermented milk inoculated with Lactobacillus brevis NZ4 and Kluyveromyces marxianus SY11 (MFM), the dominant microorganism isolated from traditional dairy products in western Sichuan.  Generally, the manuscript is well written. This paper has several weaknesses and needs improvement before publication.

This manuscript has major language problems. There are too many for me to modify them all. Authors are strongly encouraged to seek a native English speaker who may assist you modifying the document.

Comments:

1. The abstract is not particularly informative and would benefit from more background.

2. Summarize the abstract, focus on the main findings and mention the small conclusion in at the end of abstract

3. In the Introduction focus on the objectives and insert a few new reference and relevant findings

4. In material and method sections, references are missing.

5. Most of the references mentioned are old and I suggest adding recent references, and the manuscript should be edited accordingly.

6. I suggest the cite following paper in introduction part for more information you can read below reference

Jia, H., Wu, Z., Tan, J., Wu, S., Yang, C., Raza, S.H.A., Wang, M., Song, G., Shi, Y., Zan, L. and Yang, W., 2023. Lnc-TRTMFS promotes milk fat synthesis via the miR-132x/RAI14/mTOR pathway in BMECs. Journal of Animal Science, 101, p.skad218.

7. Material and method need to clarifying and summarizing- some detailed need

The subtitles in the material and method needs to summarize Ethical approval and references must be mentioned in M&M

In conclusion, the research presented is interesting, well-planned, and carried out. The manuscript can still be improved and revised by a native English speaker. Nevertheless, I believe that this work deserves publication after the inclusion of corrections.

Author Response

Please refer to appendix.

Reviewer 4 Report

Dear Authors

The subject of the study is very specific, and a very traditional practice is mentioned. Unfortunately, it is difficult to understand. It would be useful to talk about what the product is good for, how it can affect health and how it will be disseminated. The results of the study do not go beyond estimation. The results should be explained more clearly and clearly. It should be made easy to understand. This inclusivity ensures that readers without prior knowledge of the subject can not understand the concepts.

Author Response

Please refer to appendix.

Round 2

Reviewer 4 Report

Accept in present form